# Stem Cell Origin of Cancer: Implications of Oncogenesis Recapitulating Embryogenesis in Cancer Care

**DOI:** 10.3390/cancers15092516

**Published:** 2023-04-27

**Authors:** Shi-Ming Tu, Ahmet Murat Aydin, Sanjay Maraboyina, Zhongning Chen, Sunny Singh, Neriman Gokden, Timothy Langford

**Affiliations:** 1Division of Hematology and Oncology, University of Arkansas for Medical Sciences, Little Rock, AR 72205, USA; 2Department of Urology, University of Arkansas for Medical Sciences, Little Rock, AR 72205, USA; 3Department of Radiation Oncology, University of Arkansas for Medical Sciences, Little Rock, AR 72205, USA; 4Department of Pathology, University of Arkansas for Medical Sciences, Little Rock, AR 72205, USA

**Keywords:** ontogeny, oncogenesis, embryogenesis, cancer stem cells, neoantigens, microenvironment, epithelial-mesenchymal transition

## Abstract

**Simple Summary:**

Often enough, revelation of an aberrant developmental process coincides with manifestation of the malignant process. This is the essence of oncology recapturing ontogeny. This is the epiphany of tumorigenesis recapitulating embryogenesis. When embryonic processes resurface and resurge in an adult person, something is amiss. It is a telltale sign and an unmistakable hallmark of cancer. In many instances, it is a clue to the malignant tumors’ stem-cell origins. When a developmental milestone is being improperly and inopportunely expressed in a fully grown individual, something is awry. This pivotal observation is fundamental to the idea that cancer is a stem-cell disease. It is key to unlocking the stem-cell and a unified theory of cancers.

**Abstract:**

From this perspective, we wonder about the clinical implications of oncology recapturing ontogeny in the contexts of neoantigens, tumor biomarkers, and cancer targets. We ponder about the biological ramifications of finding remnants of mini-organs and residuals of tiny embryos in some tumors. We reminisce about classical experiments showing that the embryonic microenvironment possesses antitumorigenic properties. Ironically, a stem-ness niche—in the wrong place at the wrong time—is also an onco-niche. We marvel at the paradox of TGF-beta both as a tumor suppressor and a tumor promoter. We query about the dualism of EMT as a stem-ness trait engaged in both normal development and abnormal disease states, including various cancers. It is uncanny that during fetal development, proto-oncogenes wax, while tumor-suppressor genes wane. Similarly, during cancer development, proto-oncogenes awaken, while tumor-suppressor genes slumber. Importantly, targeting stem-like pathways has therapeutic implications because stem-ness may be the true driver, if not engine, of the malignant process. Furthermore, anti-stem-like activity elicits anti-cancer effects for a variety of cancers because stem-ness features may be a universal property of cancer. When a fetus survives and thrives despite immune surveillance and all the restraints of nature and the constraints of its niche, it is a perfect baby. Similarly, when a neoplasm survives and thrives in an otherwise healthy and immune-competent host, is it a perfect tumor? Therefore, a pertinent narrative of cancer depends on a proper perspective of cancer. If malignant cells are derived from stem cells, and both cells are intrinsically RB1 negative and TP53 null, do the absence of RB1 and loss of TP53 really matter in this whole narrative and an entirely different perspective of cancer?


*The more perfect a thing is, the more susceptible to good and bad treatment it is.*
Dante Alighieri

## 1. Introduction

For a malignant tumor to form, it seems that there is a master plan and an ulterior motive. The genetic defects, the epigenetic aberrations, the intercellular miscommunications, all conspire to produce a grotesque entity in the wrong place at the wrong time. Although a malignant tumor seems deformed, deranged, and destructive, it also appears to be perfect in its own ways and methods to be so dominant, resilient, and indestructible.

Likewise, for a healthy baby to be born, it seems that everything needs to be just about right. The right organs need to function in the right place at the right time. The correct tissues in the organs need to be in their proper orientations and locations. The shapes may not be consummate, the symmetry may not be immaculate, but it is hard to envision, design, or create a better magnum opus, such as a newborn baby (Figure 1). 

From this perspective, we wonder about the biological ramifications of finding remnants of mini-organs and residuals of tiny embryos in some tumors. We ponder about the clinical implications of oncology recapturing ontogeny in the contexts of neoantigens, tumor biomarkers, and cancer targets. When a tumor survives and thrives despite immune surveillance and all the restraints of nature and the constraints of its niche, it is a perfect tumor.

We speculate that a malignant tumor seems to have learned all the tricks and trades of a developing fetus. We elaborate that the current concepts of a stem-cell theory of cancer [1,2] reiterates the conventional ideas of tumorigenesis recapitulating embryogenesis. We demonstrate that a proper cancer theory on a stem-cell versus genetic origin of cancers [1,2] entails pivotal implications in cancer research and for cancer care.

## 2. Oncology Recaptures Ontogeny

When embryonic processes resurface and resurge in an adult person, something is amiss. Embryonic properties belong to an embryo, not an adult. When something that is not supposed to happen but happens anyways at the wrong place and at the wrong time, we have a nascent cancer. This pivotal observation is key to unlocking a stem-cell theory of cancer. It is fundamental to the idea that cancer is a stem-cell disease [1,2].

When it concerns embryonic events, we are observing developmental milestones being improperly and inopportunely expressed in a fully grown individual. We are not imagining that an elderly person is reversing the aging processes or activating any anti-aging mechanisms. In many respects, this is a telltale sign and an unmistakable hallmark of cancer. In many instances, it is a clue to the malignant tumors’ stem-cell origins [1,2,3,4,5,6]. 

For example, alpha-fetoprotein (AFP) is a glycoprotein produced by the yolk sac and in the liver during fetal life. Neonates have markedly elevated serum AFP levels (>100,000 ng/mL) that rapidly fall to below 100 ng/mL by 150 days and gradually change to normal (<5 ng/mL) during the first year after birth [7].

AFP is thought to be the fetal counterpart of the adult albumin. Interestingly, the AFP and albumin genes are tandemly positioned in the same transcriptional orientation on chromosome 4 [8].

The reemergence of embryonic tissues that express or produce AFP in a grown or adult person is abnormal. It suggests that a malignant process is ongoing and unfolding. In fact, the expression of AFP is often associated with germ cell tumor (such as yolk sac tumor) and with hepatocellular carcinoma [9,10]. Often enough, revelation of an aberrant developmental process coincides with manifestation of the malignant process. This is the essence of tumorigenesis recapitulating embryogenesis. This is the meaning of oncology recapturing ontogeny.

## 3. Mini-Organs

It is well known that cancer tends to be diverse and mixed, rather than simple and pure. In some ways, this general yet remarkable property of cancer, namely, heterogeneity, may be the most obvious proof if not the best evidence for a stem cell versus genetic origin of cancers.

The genetic theory of cancer stipulates genetic mutations to account for heterogeneity in cancer. In contrast, the stem-cell theory of cancer hypothesizes that aberrant stem cells with or without genetic mutations are the source, if not cause, of heterogeneity in cancer. The latter theory predicts that those same genetic defects occurring in a differentiated cell will be nonmalignant, and in a progenitor cell with less stem-ness features, less malignant with less heterogeneity [11,12].

Tata et al. [13] reported that rudimentary “stomachs, small intestines, and duodenums” were not merely present, but also seemed to be active within malignant lung tumors. 

Not unexpectedly, pathologists have reported not only mini-organs in certain tumors, but also mini-embryos in certain germ cell tumors, i.e., “polyembryomas” in embryonal carcinomas. Although embryonal carcinoma is exquisitely chemo-sensitive, “polyembryoma” tends to be chemo-resistant [14].

Perhaps we forget that, during development, the esophagus and the lung have a common origin from the endoderm. We need to remind ourselves that an aberrant pluripotent stem cell has the potential to form malignant cells with esophageal, lung, and other endoderm-derived tissue types or tumor phenotypes. 

After all, the same cancer genome in varied cells within a tumor is still fixed, rather than fluid, when a mini-lung transforms into a miniature esophagus or vice versa because of their common clonal origin.

Importantly, results from Tata’s studies support a stem cell theory of cancers, wherein a common clonal origin and a similar genetic lineage could account for tumor heterogeneity [13]. It could also explain why both lung and esophageal cancers are relatively drug resistant compared with germ cell tumors, as well as why a germ cell tumor with mini-embryos or mini-organs comprising lung or esophageal phenotype (i.e., by way of somatic transformation) becomes just as drug resistant as its lung and esophageal cancer counterparts [15].

## 4. GBM and Embryogenesis

It is uncanny that tumorigenesis embodies embryogenesis and embryogenesis epitomizes tumorigenesis. Different populations of cells within a malignant tumor have a role and function not unlike its normal analogues during embryonic development. The tumor types tend to contain diverse tissue types—a wide variety of cells that challenge and complicate treatments.

For example, glioblastoma multiforme (GBM) takes full advantage of those same processes that the nascent brain normally utilizes to create neurons and non-neurons during embryonic development. It is more than capable of repurposing or reprogramming mechanisms of invasion and seeding throughout the brain that immature neurons and neural progenitor cells are designed and destined to do during early development. It retains the capacity to form and connect with a multitude of cell types. It is essentially immune privileged and relatively drug resistant, not unlike those very immature neurons and neural progenitor cells for the same reasons and purposes.

Venkataramani et al. [16] demonstrated that GBM cells form a network with one another, connected by long protrusions known as tumor microtubes (TM). When neural stem cells and cancer stem cells have a common developmental origin and share similar stem-ness properties, perhaps it should not be a surprise that neuronal signals fuel the growth of TM in both developing brains and GBMs. Similarly, Xie et al. [17] showed that gap junction-coupled tumor cells in TM-connected glioma networks, as compared with unconnected tumor cells, correlated with increased cellular stemness. 

Perhaps the TM story may have solved an enduring mystery of GBM, in which certain genetic aberrations turn out to be fortuitous rather than deleterious in certain tumors, i.e., 1p–19q co-deleted oligodendrogliomas entail a better prognosis than 1p–19q intact astrocytomas [18]. Hence, molecular factors localized on chromosomal arms 1p and 19q, which are co-deleted in oligodendroglioma, involve genes encoding TTYH1 that drives invasive TM, as well as those encoding the neurotrophic factors nerve growth factor and neurotrophin 4, which increase expression of GAP43, a master regulator of TM formation. This explanation accounts for the low expression levels of TTYH1 and GAP43 in oligodendrogliomas compared with those in astrocytomas.

Importantly, how cellular networks drive malignancy in aggressive human brain tumor types has pertinent therapeutic implications. For example, meclofenamate (MFA), a non-steroidal anti-inflammatory drug, is a gap junction inhibitor. MFA has been shown to reprogram the developmental profiles towards a less connected cellular state [19]. It may provide a favorable therapeutic window by targeting the fraction of highly connected glioblastoma cells while concomitantly sparing the fraction of less connected adult physiological brain cells in which developmental programs are either limited or halted. MFA may be particularly beneficial for the purposes of network-targeting intervention in adjuvant or maintenance therapy of GBM.

## 5. The Perfect Tumor

People may not realize that the fetus is a perfect tumor in many respects and aspects. Fortunately, it is almost never malignant. When there happens to be a genetic defect that affects its viability, it may either surreptitiously disappear or spontaneously abort itself. Perhaps if a fetus is not born, it will become or is practically malignant. Fortunately, nature has its own ways and means to take care of itself.

The fetus grows at a prolific rate, just as some tumors do. In fact, it grows faster than most cancers. Before a baby is born, it protrudes in the mother’s abdomen like a massive tumor does. At nine months, the average length and weight of a newborn is 19.5 inches and 7.25 pounds, respectively [20]. Very rarely do we allow a tumor to grow to this size in our body.

Because a fetus sucks up all the nutrients from its mother and hardly gives anything back in return, it behaves almost similar to a parasite. It seems to adversely affect the good health and wellbeing of the mother without killing her. Somehow, it found a way to survive and thrive in another body that will normally never accept it and will most certainly reject anything other than her own in her body.

After all, part of the baby is from the father who can be as different from the mother as he can be. However, the mother accepts a baby who may be half her own self, unconditionally. Indeed, even in a surrogate pregnancy, the “mother” will accept what is 100% not hers, unequivocally. 

What enables a fetus to evade and elude the mother’s immune surveillance and what protects the fetus from a mother’s supposedly intact and competent immune system could be the very same processes and mechanisms that enable and protect a malignant tumor from its host. 

The baby that is born, and manages to form similarly to a tumor, is a success story in its own rights by all measures. 

We are elated to witness the birth of a perfect baby. We are amazed to observe the rise of a perfect tumor.

## 6. Neoantigens

The concept of neoantigens is ingenious—or ill-conceived—depending on our perspective of cancer [21]. If it is based on pertinent clinical observations and valid scientific hypotheses, then it is pivotal and can be paradigm shifting in our practice of cancer care. However, if it is mainly based on experimental evidence that generates, rather than tests, a specific scientific hypothesis, then it is inherently flawed and could be self-serving and self-fulfilling. It may be right for the wrong reasons [22].

The very idea of neoantigens is problematic in the context of a so-called perfect tumor—the fetus. In a mother who is otherwise young and healthy with an intact and robust immune system, the fetus is spared despite its being half foreign and bearing an abundance of neoantigens derived from its father. It grows similarly to a tumor and dodges the mother’s immune surveillance similar to how a cancer does. It is conceivable and perhaps inevitable that a malignant tumor takes advantage of the same ploys to preserve and protect itself from the ravages of the immune system similar to how a fetus does in a mother despite its alienness and its neoantigens. 

The relevance of neoantigens is also ignored and often neglected in the setting of stemness when a progenitor cell downregulates expression of MHC type I molecules and when those neoantigens would be tolerated as self rather than eliminated as non-self [23,24,25]. Again, cellular context is paramount because, when a progenitor cell differentiates into a progeny cell and upregulates MHC type I and II molecules, the immune cells are more empowered to recognize and eradicate a defective progeny cell with neoantigens. We can afford to eliminate progeny cells, rather than progenitor cells, since we can always replenish the former with the latter. Otherwise, we prevent malignancy, but promote senescence—we activate anticancer mechanisms at the expense of anti-aging processes [26,27]. 

Bohnert et al. [28] and Voutsadakis [29] demonstrated that proteostasis and its counterpart, proteolysis, affect the presence of neoantigens in a progenitor cell versus progeny cell. After all, it is critical to keep an indispensable progenitor cell pristine and impeccable—unlike a progeny cell, which is dispensable and replaceable. Progenitor stem cells are proficient in DNA repair. They are also seasoned in protein repair through autophagy and senescence. We cannot afford to keep and pass defective DNA and proteins to our offspring. Contrary to conventional wisdom, this natural housekeeping ensures that there will be much fewer neoantigens in a progenitor than in a progeny cell. Therefore, if our goal is to control differentiated cancer cells, targeting neoantigens makes sense. However, if we aim to counter cancer stem cells, targeting neoantigens is less likely to be effective.

Again, when we think of neoantigens we need to consider cellular context versus genetic content, which is closely intertwined with a stem-cell versus genetic origin of cancer [30]. What role do neoantigens play in aneuploidy, in which there is an abnormal number of chromosomes in a malignant cell? According to Klein et al. [31], aneuploidy correlates with fewer genetic mutations, i.e., there is abnormality in gene dosages rather than in gene defects. When there is an abundance or dearth of native gene products, rather than an appearance of novel genetic entities, the problem is with the presence or absence of abnormal self-antigens, rather than the formation of neoantigens in aneuploidy. Importantly, aneuploidy is closely associated with defective asymmetric division, which is an essential stemness property. Again, a stem cell versus genetic origin of cancer predicts that neoantigens would be less prevalent in tumors derived from progenitor stem cells compared with those from progeny-differentiated cells.

Therefore, the very idea of a stem cell versus genetic origin of cancer suggests that the concept of neoantigens may be flawed. When we formulate an erroneous hypothesis based on faulty observations of nature regarding neoantigens, we may still perform exemplary experiments to test the false hypothesis and obtain convincing results to support the hypothesis [22,32]. However, the clinical reward, if any, derived from such “preclinical” and “translational” research, could be haphazard, erratic, and academic, and the clinical benefits, at best, would be coincidental, incremental, and marginal. 

## 7. Embryonic Niche

In an onco-niche, one can imagine that cancer stem cells and accessory cells (such as inflammatory cells) are in full bustle and hustle. There is a mob and a riot. It is similar to a wound that does not heal.

In an embryonic niche, one can also envision that normal stem cells and ancillary cells are sprouting and blossoming. However, there is order in disorder. There is a start and a stop.

In a thriving community, both the neighbors and neighborhood support one another. In a crumbling society, the residents plunder the environment, and the environment corrupts the residents [33].

Results from classical experiments suggest that the embryonic microenvironment possesses antitumorigenic properties. For example, Stevens demonstrated that stem cells arising from the gonadal ridge of an embryo converted to an embryonal carcinoma when grafted onto an adult mouse testis [34]. Conversely, Mintz et al. [35] showed that embryonal carcinoma cells inserted into the inner cell mass of a mouse blastocyst behaved similar to normal stem cells and became part of a “mosaic” mouse without overt cancer. Intriguingly, Rous sarcoma virus did not induce sarcomas in chicken embryos [36]. B16 murine melanoma cells failed to form tumors after exposure to niche factors derived from the embryonic mouse skin [37].

During early embryonic development, there is a dearth of immune cells because the embryo’s immune system has not yet matured. Experimentally, this is an ideal condition for the implantation of tumor cells onto the chicken chorioallantoic membrane with minimal risk of xenograft rejection [38]. Theoretically, when the embryonic immune system matures and becomes competent, it should recognize those alien neoantigens and eliminate the xenograft. The observation that a physiological, immune-reactive, in vivo model system does not reject a xenograft, which can still be conveniently manipulated for the purposes of research, is reminiscent of an immune competent mother impregnated with a baby and patient inflicted by cancer. It implores that we need to formulate a more pertinent theory about immune surveillance and tolerance regarding “neoantigens”, rather than continue to embrace one that is orthodox, but may be obsolete.

Therefore, an elemental, if not existential, question relates to the nature of embryonic cells as much as to the immune cells themselves. It is plausible that those tumors derived from indigenous embryonic and progenitor stem cells are similarly immune privileged and protected, similar to their foreign exogenous counterparts. In an embryonic niche, there is no boundary between self and non-self. There is also a blur whether the immune system is initially immature, but appropriately incompetent, or subsequently matures and becomes incompletely competent.

## 8. Stem-Ness Niche

It is conceivable that niche signals produced by host stem cells and differentiated cells, fibroblasts, macrophages, T cells, platelets, et cetera, in a multicellular system affect the function and nature of both individual cells and the whole system.

Specifically, co-localization of regulatory T cells (Treg) with adult stem cells (ASC), such as hematopoietic stem cells (HSC) and hair follicle stem cells, form unique stem-ness niches in which they generate an immune-privileged microenvironment that protect and preserve normal stem cells and cancer stem cells (CSC) [39,40]. In many respects, this is both necessary and mandatory because we cannot afford to have our immune system destroy our own normal stem cells.

According to a stem cell theory of cancer, we predict that our regular immune system is incapable of eradicating a malignant tumor that is well endowed with stemness properties either. We anticipate that, even though immunotherapy is effective for the treatment of tumors branded or tagged for elimination by the immune system, it may be less effective for the treatment, let alone for the cure, of a preponderance of those cancers well equipped with a multitude of stem-like and immune evasive capabilities, including heterogeneity, dormancy, drug resistance, and immune tolerance.

After all, CSC resembles embryonic stem cells (ESC) and ASC. Tipnis et al. [41] demonstrated that CSC and normal stem cells have a common lineage and share a close residence with more stem-ness features—increased Treg, as well as presence of anti-inflammatory macrophages and mesenchymal stem cells (MSC), which promote immunosuppressive responses through increased IL-10, IDO, and PGE2 that harbor and favor a vast majority of high-grade cancers. Zhou et al. [42] showed that activated immune cells, such as T cells and NK cells, produce gamma IFN, which decreases MHC expression and impairs differentiation of stem cells. An inadvertent consequence of this ESC-ASC/MSC axis and stem-ness niche is immune privilege and immune evasion of CSC.

However, when stem cells move down a stem-cell hierarchy of progenitor and progeny cells, they mobilize to their respective stem-ness and stem-less niches, respectively. For HSC, this means moving away from the endosteal to the sinusoid niches [43]. In the latter locales, there are fewer Tregs and more dendritic cells and antigen presenting cells. Even the MSC and macrophages assume altered roles and converted functions—they are now pro-inflammatory, rather than anti-inflammatory, and promote immune-stimulatory, rather than immune-inhibitory, responses through increased IL-6, IL-8, TNF-alpha, TGF-beta, and similar components, which enhance MHC expression and enable differentiation of stem cells [44,45,46].

It is ironic that, in both normal and malignant stem-ness niches, some of the crucial cells involved (e.g., MSC and macrophages) and the cytokines present (e.g., TNF-alpha and TGF-beta) stay in a delicate balance of quiescence and activation with dual roles and shifting functions [47]. That is, they can be both pro-inflammatory or anti-inflammatory, immune-activating or immune-suppressive, depending on an obligate interplay between their inherent nature and integrated niches. In other words, they can be both the producer and product of a specific stem-ness marker and malignant target.

## 9. Paradox of TGF-Beta

A paradox in cancer biology relates to an entity that is supposed to be good but can also be bad. How can it be that TGF-beta is both a tumor suppressor and promoter [48,49,50]? This Dr. Jekyll and Mr. Hyde-tendency of TGF-beta is emblematic and problematic in cancer. To understand this paradox is to elucidate the origin and nature of cancer.

TGF-beta is a tumor suppressor because it mediates anti-proliferative and apoptotic effects. It is a vital stromal factor affecting cellular differentiation and mobilization in many kinds of tissues, including the bone and lungs. 

TGF-beta is also a tumor promoter because it induces tumor motility, invasion, metastasis, and epithelial-to-mesenchymal transition (EMT). The “switch” of TGF-beta from suppressor to promoter seems to occur in some form or another during malignant transformation in all cancers.

However, a narrative of cancer depends on our perspective of cancer. According to the genetic theory of cancer, TGF-beta is king. When we focus on TGF-beta in a reductionist view of cancer, it determines and dictates cancer. TGF-beta makes or breaks cancer. However, according to a stem cell theory of cancer, TGF-beta is a pawn. In an integrated view of cancer, TGF-beta is but one of many pieces in a game of chess and various components in a system. Importantly, we propose that a proper theory about the origin of cancer has pertinent therapeutic implications, i.e., on drug versus therapy development [15]. Otherwise, a highly touted TGF-beta-targeted therapy, bintrafusp alfa, may be successful for the purposes of drug development, but it becomes another abject failure for the purposes of therapy development in cancer care [51].

Therefore, it is imperative for us to realize, if not to know, that TGF-beta is a different actor and player in the fetus versus an adult, during embryogenesis versus carcinogenesis. What it does in the fetus may be perfectly benign during embryogenesis, but, in an adult, the same action and activity may be patently malignant during carcinogenesis. 

Again, this is another instance of oncology recapturing ontogeny. When embryogenic factors reemerge and resurge, tumorigenic processes manifest and manipulate. The paradox of TGF-beta as a tumor suppressor and promoter is also a key to unlocking the stem cell theory of cancer. When cancer has a stem-cell origin and is a stem-cell disease, it does not need to reprogram or reinvent itself, it does not need to hijack what it already owns or negate what it does not owe.

## 10. Dualism in EMT

EMT was first recognized as a feature of embryogenesis in limb regeneration [52,53]. Subsequently, EMT was demonstrated to play a role in the process of gastrulation (reorganization of the single-layered mass, blastula, into a multilayered structure, gastrula), the formation of neural crest (which gives rise to diverse cell lineages, including melanocytes, peripheral neurons, and glial cells), and many other developmental events in the embryo [54].

Coincidentally, EMT is also a stem-ness trait involved in both normal development and certain disease states, including cancer [54,55,56,57]. Notably, several investigators have demonstrated that embryogenesis shares the same stem-ness genes (such as N-cadherin, SNAIL, ZEB) and employs the same EMT pathways (such as WNT/beta catenin, Hedgehog) as carcinogenesis does [58,59,60].

EMT and its reverse mesenchymal-to-epithelial transition (MET) are integral to organogenesis and to oncogenesis. Both implicate if not indicate that a multipotent cell has the capacity to evolve into at least two separate and distinct cellular lineages with an epithelial and/or mesenchymal phenotype.

According to a stem cell origin of cancer, EMT and MET are mere manifestations of stem-ness properties: self-renewal, pluripotency, and differentiation [1,2]. However, according to another version of a stem cell theory of cancer, they provide evidence for a contrary concept of stem-ness properties, namely, bi-directionality in the stem cell hierarchy (in contrast to uni-directionality), in which dedifferentiation, transdifferentiation, and reprogramming may also be at play during embryogenesis and oncogenesis.

Therefore, EMT is another developmental landmark that is also a cancer hallmark. It alludes to dualism in a basic biological and pathological process. It exemplifies a recurring theme that a similar activity or even identical action may be benign in an embryonic body but malignant in an adult person.

## 11. Chicken or Egg

When we adopt the genetic theory of cancer, we postulate that defective p53 and BRAF are involved in the formation of melanoma, and p53 and RB1 loss are responsible for the development of small cell carcinoma. However, when we advocate a stem-cell theory of cancer, we perceive the same genetic defects in a different light and prescribe them a different role. 

Importantly, Kaufman et al. [61] demonstrated that no cell with defective p53 and BRAF becomes a melanoma, unless the cell also happens to be a crestin-expressing stem cell [52]. Furthermore, Mu et al. [62] showed that small cell carcinoma with p53 and RB1 loss reverses its phenotype when it stops expressing a stem-ness gene, namely, SOX2. 

Assume that a progenitor stem cell has always had inactive tumor suppressor genes (such as p53, RB1) because that is the innate constitution of stem cells [63]. Then, what is the significance of a negative RB1 and null TP53 in a malignant melanoma or small cell carcinoma derived from a cancer-initiating progenitor stem cell? 

Remember that, during fetal development, proto-oncogenes wax, while tumor-suppressor genes wane. Similarly, during cancer development, proto-oncogenes awaken, while tumor-suppressor genes slumber. In stem cells, self-renewal genes, such as *bmi-1,* shut down the tumor-suppressor gene, *p16^INK4a^* [63]. Therefore, stem cells could be naturally or virtually RB1 negative and TP53 null. 

If malignant cells are derived from stem cells, and both cells are inherently RB1 negative and TP53 null, does RB1 and TP53 loss really matter in this entirely different perspective and an alternative perhaps unconventional narrative of cancer? 

That depends on which and what theory we subscribe to regarding the origin of cancer. When we believe in the genetic theory of cancer, defective p53 and BRAF cause melanoma, and p53 and RB1 loss cause small cell carcinoma. However, if we believe in a stem-cell theory of cancer, aberrant progenitor stem cells with the same or different genetic defects become malignant, while aberrant progeny non-stem cells with the same or different genetic defects do not/may not.

In other words, a benign mole in the skin with defective p53 and BRAF may never become a melanoma because it is not derived from a stem cell. One would predict that benign prostatic hyperplasia with p53 and RB1 loss will not convert into prostate cancer, either, for the same reasons. 

In the genetic theory of cancer, we do not know if TP53 is the chicken or the egg. In a stem-cell theory of cancer, it may not matter if TP53 is the chicken or the egg. 

## 12. Therapeutic vs. Theoretic

It is important to separate practice from principles and reality from theories in cancer care. However, ideas and actions are often inextricably linked: what we believe affects what we do. Different cancer theories will lead to distinct research directions and treatment developments with disparate clinical implications.

Perhaps it is not mere coincidence that anti-stem-ness activity has anti-cancer effects because of an intrinsic, yet intricate, parallel, but duplicate, interplay between embryogenesis and oncogenesis, as well as between stem cell and cancer biology.

### 12.1. Trop2

Trop2 was first isolated from monoclonal antibodies generated against a human choriocarcinoma cell line BeWo, as well as from nonmalignant trophoblast cells [64]. It is highly expressed in stem cells within various organs during embryogenesis, and in a variety of human malignancies, including triple negative breast cancer (TNBC) and upper tract urothelial carcinoma. Over-expression of Trop2 in human tumors promotes tumor aggressiveness and increases patient mortality [65,66]. 

Trop2’s multifaceted role includes regulation of cell proliferation and migration, self-renewal, and maintenance of basement membrane integrity. It enhances stem-like properties of cancer cells through beta-catenin signaling [67].

Sacituzumab-govitecan is an effective cancer therapeutic that targets Trop2. It is approved for the treatment of refractory TNBC and metastatic urothelial carcinoma [68,69].

### 12.2. Nectin4

Among the four human nectins, nectin4 is unique in that its expression is largely restricted to placental and embryonic tissues. In contrast to healthy adult tissue, many cancer types, including breast, ovarian, cervical, colorectal, esophageal, gastric, lung, liver, and thyroid cancers, have high nectin4 expression. High nectin4 expression is associated with increased tumor size, grade, and invasiveness, as well as reduced patient survival [70]. 

Nectin4 is a stem-cell biomarker that upregulates EMT and metastasis. It induces WNT/beta-catenin signaling via the Pi3k/Akt axis [71]. It cooperatively regulates with p95-ErbB2 Hippo signaling-dependent *SOX2* expression [72].

Enfortumab is an effective cancer therapeutic that targets nectin4. It is approved for the treatment of locally advanced or metastatic bladder cancer [73].

### 12.3. RTK

Receptor tyrosine kinases (RTK) are ubiquitous stem-ness factors that play a vital role in diverse embryonic and malignant processes, including pluripotency/differentiation, self-renewal/cell fate, morphogenesis, migration/invasion, et cetera [74,75]. 

Tyrosine kinase inhibitors (TKIs) are known to be teratogenic, in part because they disrupt embryogenesis. Not unexpectedly, because TKIs also interfere with oncogenesis they can be utilized for therapeutic purposes in cancer care.

For example, HER-2 is innately connected with stemness. It interacts with IL-8R (CXCR1/2) in the regulation of breast CSC by the tumor microenvironment [76,77].

Several HER-2 targeted therapies are effective and have been approved for the treatment of HER-2+ breast and gastric cancers [78].

## 13. Conclusions

Vladimir Horowitz, an American-Russian classical pianist said that “perfection itself is imperfection”. 

In a perfect baby, progenitor stem cells and their specialized progeny cells are more than well ordained, organized, and orchestrated. In a perfect tumor, these same cells become untethered, unmoored, and unleashed.

Unfortunately, a perfect baby is vulnerable to defects, deformity, and disease. Perhaps a perfect tumor made in its own embryonic image is also susceptible to both “good and bad treatments”.

In many respects, oncology recapturing ontogeny is a distorted mirror image of tumorigenesis and embryogenesis, a caricature of CSC in ESC/ASC (Figure 2). It is reflected in mini-organs and microtubes. It is parodied in neoantigens. It is immersed in niches, but it emerges from EMTs.

When we manage to improve the clinical outcome of more cancer patients and increase the cure rate of a minority of those patients with current therapeutic modalities, perhaps the treatment is effective for the wrong reasons. When we have the right cancer theory and we learn how to harness the power and benefit of those same treatments for the right reasons, perhaps the treatment will benefit more patients with cancer in more substantial than subliminal ways. 

We propose that good treatment of cancer subscribes to a comprehensive, unified stem-cell theory of cancer that considers its embryonic seeds and soil, as well as its developmental roots and tree in the prevention of its mistakes and miscues, treatment of its multiple compartments and components, and maintenance of any durable remissions. Bad treatment only submits to a small or limited part of a genetic origin of cancer that often ignores its prevention, neglects its complexity, and dismisses its perpetuity.

## Figures and Tables

**Figure 1 cancers-15-02516-f001:**
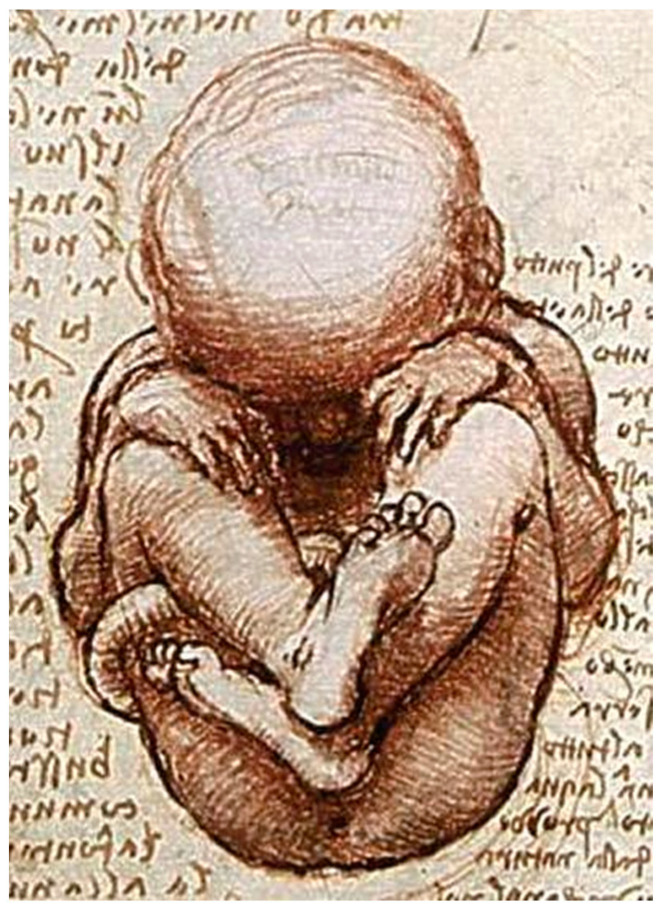
A fetus in the womb, by Leonardo da Vinci, circa 1510. https://commons.wikimedia.org/wiki/File:Views_of_a_Foetus_in_the_Womb_detail.jpg (accessed on 28 January 2023).

**Figure 2 cancers-15-02516-f002:**
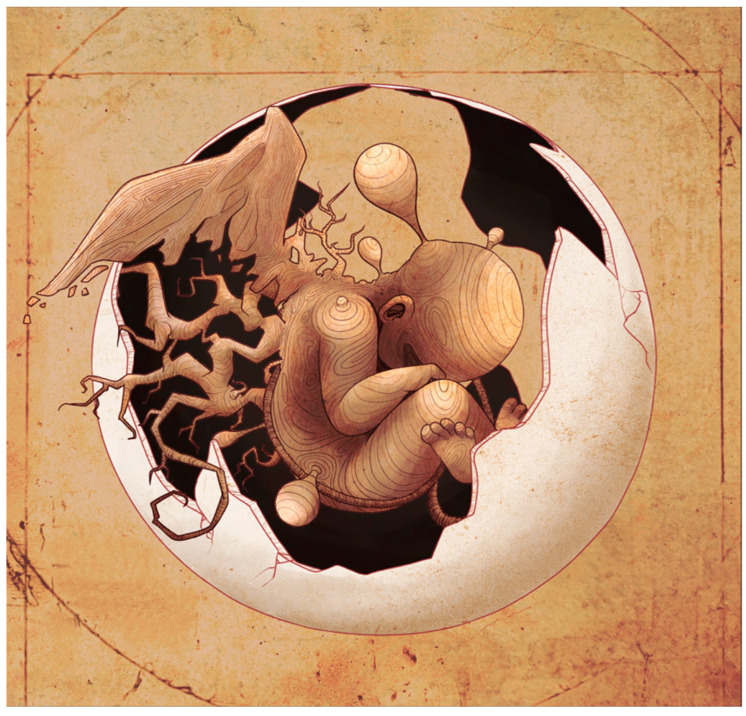
An artist’s rendition of oncogenesis recapitulating embryogenesis in a perfect tumor. Reproduced with permission from Benjamin Tu.

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
