# Peer review of "Stem Cell Origin of Cancer: Implications of Oncogenesis Recapitulating Embryogenesis in Cancer Care"

_cancers, 2023, doi:10.3390/cancers15092516_

Round 1
Reviewer 1 Report
This Perspective provides a discussion of the stem cell origin theory of cancer as it relates to embryonic processes. Although there are some potentially new interesting and valid points, there are several issues that should be clarified.
It is not clear what is meant by the “stem-cell theory of cancer”; the way it is presented it implies that this is a new theory that the authors have previously proposed and are further arguing for. However, the stem cell origin of cancer theory has been around for some time now. Nevertheless, in this perspective there is no reference to previous proposals as to the role of stem cells in the development of cancer and the implications of this aspect; and no discussion of how the ideas in this perspective (or the authors’ version of the “theory”) relate to previous treatments of the role of stem cells in other cancer theories. The latter include ideas about the reactivation of embryonic pathways – including the suggestion that aspects of the cancer cells that reflect stages in fetal development can be used as markers for diagnosis, prognosis and treatment (onco-developmental markers) (see for instance, Sell 2010; Aiello and Stanger 2016; Kim et al. 2017; but also many others). Are the authors suggesting something different? Is their statement that “cancer is a stem-cell disease” what they want to argue for in this perspective? A better discussion of how their views differ from previous implications of stem cells in oncogenesis would benefit the reader. Also, citing some of the work that proposed the stem-cell origin of cancer is necessary (for instance, the statement in line 48 – “the malignant tumors’s stem cell origin”, should include a reference). The authors seem to reference their previous papers on a “stem-cell theory of cancer”, but that theory (and how it differs from other theories) is not described/discussed. Although on lines 335-340 they distinguish between a “stem cell origin of cancer” theory (for which they reference 2 of their papers) and “another version of a stem cell theory of cancer” for which they do not provide any references… Again, this is very confusing and needs clarification….
References are needed for the statements about AFP (lines 57-60). Also, the connection between expression of AFP in teratocarcinoma and expression of embryonic traits has already been reported. This connection is the only example/argument used in the section “Oncology recapitulates ontogeny”….
The use of ‘oncology’ in phrases such as ‘oncology recapitulates ontogeny’ does not seem appropriate as ‘oncology’ refers to a science or branch of medicine, not a process; ‘oncogenesis’ is a better term.
Not sure if the statement “oncology recapitulates ontogeny” is really appropriate; ‘recapitulates’ is usually used to mean that the series of events is ordered in the same way/succession… See the concept of “ontogeny recapitulates phylogeny”… Although there might be aspects of ontogeny that are reactivated during oncogenesis, I do not think one can state (especially in the title) that oncogenesis ‘recapitulates’ embryogenesis – at least not in all tumors. It seems that what the authors are arguing is that embryonic traits/genes are reactivated during cancer progression; which is not a new concept/idea.
Also, not sure if the statements “perfect tumor” and “fetus is a perfect tumor” are valid… For instance, regarding the latter, the authors acknowledge that genetic defects in a fetus are either eliminated or induce fetus abortion (lines 134-35)… that is not the case for a tumour. In fact, the authors describe tumours as being perfect in terms of their resilience – which is not the case with a fetus… Generally, the term ‘perfect’ is rather hard to define… The way in which a tumor and a baby are perfect are rather different… Furthermore, the term ‘tumour’ – by definition, denote a mass resulting from uncontrolled cell proliferation… to call a fetus a “perfect tumor” seems inappropriate taking into account the need for controlled proliferation that defines embryonic development… it does not “grow like a tumor” (line 170) as developmental processes are genetically coded and always result in a precise/defined morphology, which is not the case for tumors.
A discussion (and references) of how the immune system deals with the presence of a fetus would be useful to appreciate the points the authors make about neoantigens in tumors.
Lines 341-344: “Therefore, EMT is another developmental landmark that is also a cancer hallmark. It alludes to dualism in a basic biological and pathological process. It exemplifies a recurring theme that a similar activity or even identical action may be benign in an embryonic body but malignant in an adult person.”. This connection has been pointed before and is a well-known link between developmental processes and carcinogenesis (see for instance Kim et al 2017).
Not sure the statement “during fetal development proto-oncogenes wax, while tumor-sup pressor genes wane”; for instance, p53 is highly expressed during embryonic development and appears to also have developmental roles (see, for instance, Shin et al. Cell & Bioscience 2013, 3:42)
Lastly, taking into account the proposed similarities between cancer and embryonic development – via the proposed common origin (ie, stem cells), the authors should discuss what type of changes divert the stem cell towards a tumour instead of an embryo. The last section discussing p53 for instance, seems to negate the need for mutations in tumor suppressor genes (ie, if p53 is the “egg”).
Many statements (including specific information/data; eg, lines 117-122; 124-127; 176) lack references.
---
references referred to above
Sell S. Stem Cell Origin of Cancer, AJP June 2010, Vol. 176, No. 6. DOI: 10.2353/ajpath.2010.091064
Marongiu, F., Serra, M. and Laconi, E. (2018) Development versus evolution in cancer biology. Trends in Cancer 4, 342–348,
https://doi.org/10.1016/j.trecan.2018.03.007
Chen, H. and He, X.L. (2016) The convergent cancer evolution toward a single cellular destination. Mol. Biol. Evol. 33, 4–12, https://doi.org/10.1093/molbev/msv212
Aiello, N.M. and Stanger, B.Z. (2016) Echoes of the embryo: using the developmental biology toolkit to study cancer. Dis. Model. Mech. 9, 105–114,
https://doi.org/10.1242/dmm.023184
Kim, D.H., Xing, T., Yang, Z., Dudek, R., Lu, Q. and Chen, Y.H. (2017) Epithelial mesenchymal transition in embryonic development, tissue repair and cancer: a comprehensive overview. J. Clin. Med. 7, 1, https://doi.org/10.3390/jcm7010001
Author Response
The reviewer is right that “the stem cell theory of cancer” is not a new theory. Most people attribute this idea to Virchow in 1863 (reference 1).
It is true that an introduction to this subject may clarify the questions he asks. In fact, references 1 and 2 in the introduction of this article comprise two books with a total of about 856 references that address the history, controversies, implications, etc. related to the stem cell theory of cancer. In addition, references 11 (metastasis), 12 (heterogeneity), 14 (drug resistance), 22 (translational research), 26 (aging), and 27 (dormancy) updated specific topics pertinent to this theory with additional references (total of about 326), which this current article also does with respect to the specific topic of embryogenesis.
Thank you for suggesting the Sell, Aiello and Stanger, Kim et al and other references, which we have included in this article.
Sell S. On the stem cell origin of cancer. Am J Pathol 2010; 176:2584-94.
Marongiu F, Serra M, Laconi E. Development versus evolution in cancer biology. Trends Cancer 2018; 4:342-8.
Chen H, He X. The convergent cancer evolution toward a single cellular destination. Mol Biol Evol 2016; 33:4-12.
Aiello NM, Stanger BZ. Echoes of the embryo: using the developmental biology toolkit to study cancer. Dis Model Mech 2016; 9:105-14.
Kim DH, Xing T, Yang Z, Dudek R, Lu Q, Chen YH. Epithelial mesenchymal transition in embryonic development, tissue, repair, and cancer: a comprehensive overview. J Clin Med 2017; 7:1.
References are needed for the statements about AFP (lines 57-60). Also, the connection between expression of AFP in teratocarcinoma and expression of embryonic traits has already been reported. This connection is the only example/argument used in the section “Oncology recapitulates ontogeny”….
Done!
Shojaei H, Hong H, Redline RW. High-level expression of divergent endodermal lineage markers in gonadal and extragonadal yolk sac tumors. Modern Pathol 2016; 29:1278-88.
Galle PR, Foerster F, Kudo M, Chan SL, llovet JM, Qin S, Schelman WR, Chintharlapalli S, Abada PB, Sherman M, et al. Biology and significance of alpha-fetoprotein in hepatocellular carcinoma. Liver International 2019; 39:2214-29.
The use of ‘oncology’ in phrases such as ‘oncology recapitulates ontogeny’ does not seem appropriate as ‘oncology’ refers to a science or branch of medicine, not a process; ‘oncogenesis’ is a better term. Not sure if the statement “oncology recapitulates ontogeny” is really appropriate; ‘recapitulates’ is usually used to mean that the series of events is ordered in the same way/succession… See the concept of “ontogeny recapitulates phylogeny”… Although there might be aspects of ontogeny that are reactivated during oncogenesis, I do not think one can state (especially in the title) that oncogenesis ‘recapitulates’ embryogenesis – at least not in all tumors. It seems that what the authors are arguing is that embryonic traits/genes are reactivated during cancer progression; which is not a new concept/idea.
Agree with reviewer that “oncology recapitulates ontogeny” is a novel term that challenges current and conventional thinking. For us oncologists who diagnose and treat cancer, oncology and the practice of oncology is without doubt a process. It is true that our observation of embryonic traits reemerging and resurging during cancer progression is common, not new. However, to encapsulate this important observation so that we can enhance the practice of oncology (i.e., diagnosis and treatment of cancer) in drug versus therapy development would be innovative (section 8, paragraph 4, line 345).
Thank you for your astute comments and insightful suggestions: We emphasize “Implications” in the title of this paper and have added “in Cancer Care” to clarify the basic ideas and main purposes of this manuscript.
Also, not sure if the statements “perfect tumor” and “fetus is a perfect tumor” are valid… For instance, regarding the latter, the authors acknowledge that genetic defects in a fetus are either eliminated or induce fetus abortion (lines 134-35)… that is not the case for a tumour. In fact, the authors describe tumours as being perfect in terms of their resilience – which is not the case with a fetus…
Generally, the term ‘perfect’ is rather hard to define… The way in which a tumor and a baby are perfect are rather different… Furthermore, the term ‘tumour’ – by definition, denote a mass resulting from uncontrolled cell proliferation… to call a fetus a “perfect tumor” seems inappropriate taking into account the need for controlled proliferation that defines embryonic development… it does not “grow like a tumor” (line 170) as developmental processes are genetically coded and always result in a precise/defined morphology, which is not the case for tumors.
Although it is difficult to demonstrate elimination of defective tumors, the phenomenon of spontaneous remission suggests that it does happen (Reference 2, chapter 17). When babies are born with missing body parts or defective genes, there is some degree of resilience in their viability.
Agree that “perfect” is hard to define. When a complex and complicated process such as embryogenesis and tumorigenesis is completed, it seems to be perfect in its own ways and methods. Otherwise, there should be no newborns or cancers.
It is true that a fetus and a tumor cannot be more different, but any similarity between the two is what may be missed or has been dismissed in our thinking. We propose that a similar biological process and mechanism of action between two disparate entities such as a fetus and a tumor in a stem cell origin and within the proper cellular context may provide a novel perspective in the way we understand cancer and how we treat it.
A discussion (and references) of how the immune system deals with the presence of a fetus would be useful to appreciate the points the authors make about neoantigens in tumors.
Done!
In section 5, Neoantigens, paragraph 3, we discussed how the immune system deals with neoantigens in the context of MHC I (references 23-25), and paragraph 4, proteostasis/proteolysis in progenitor stem cells versus progeny differentiated cells in a normal fetus as well as in a malignant tumor (references 20, 21).
Lines 341-344: “Therefore, EMT is another developmental landmark that is also a cancer hallmark. It alludes to dualism in a basic biological and pathological process. It exemplifies a recurring theme that a similar activity or even identical action may be benign in an embryonic body but malignant in an adult person.”. This connection has been pointed before and is a well-known link between developmental processes and carcinogenesis (see for instance Kim et al 2017). Not sure the statement “during fetal development proto-oncogenes wax, while tumor-sup pressor genes wane”; for instance, p53 is highly expressed during embryonic development and appears to also have developmental roles (see, for instance, Shin et al. Cell & Bioscience 2013, 3:42)
Agree with reviewer about the well-known connection between developmental processes and carcinogenesis, which is the main theme of this article.
Pardal et al (reference 54) demonstrated that stem cell self-renewal and cancer cell proliferation are regulated by common networks that balance the activation of proto-oncogenes and tumor suppressors. “For example, the polycomb family proto-oncogene, Bmi-1, is consistently required for the self-renewal of diverse adult stem cells, as well as for the proliferation of cancer cells in the same tissues. Bmi-1 promotes stem cell self-renewal partly by repressing the expression of Ink4a and Arf, tumor suppressor genes that are commonly deleted in cancer.”
My understanding of Shin’s paper with regards to specific proto-oncogenes and tumor suppressor genes in a whole network of proto-oncogenes and tumor suppressor genes rather than an isolated p53 is consistent with Pardal et al’s seminal studies. “The absence of obvious developmental defects in p53 knockout mice strongly suggests that p53 is not required for development. Other studies, however, suggest that p53, under certain conditions, is involved in the development of mice.”
Lastly, taking into account the proposed similarities between cancer and embryonic development – via the proposed common origin (ie, stem cells), the authors should discuss what type of changes divert the stem cell towards a tumour instead of an embryo.
In this article, we highlight the idea that in the study and treatment of cancer in a multicellular person, we need consider cellular origin and cellular context (e.g., progenitor cell vs progeny cell), and the microenvironment (stem-like vs specialized niche). The same defect in a different cellular context and microenvironment elicits different embryonic and malignant phenotypes (section 6, Embryonic niche, references 34-37). This is the essence of tumorigenesis recapitulating embryogenesis. Otherwise, when we design a treatment that targets a specific genetic defect (e.g., TGF-beta) without considering its cellular origin/context and the microenvironment, the treatment is likely to provide only marginal and transient (if any) clinical benefit (section 8, Paradox of TGF-beta, reference 51).
The last section discussing p53 for instance, seems to negate the need for mutations in tumor suppressor genes (ie, if p53 is the “egg”). Many statements (including specific information/data; eg, lines 117-122; 124-127; 176) lack references.
The question about p53 was addressed above by referring to Pardal et al (reference 54).
Reference 18 for lines 152-157 (lines 117-122 in the prior version) was already in place. Apologize that this was not apparent despite use of the word “Hence” following the reference to connect the reference with the sentence.
Similarly, reference 19 for lines 159-161 (lines 124-127 in the prior version) was already in place after the following sentence (prior line 130). We have moved this reference up one sentence if the reviewer thinks that by doing so would enhance clarity of the reference already provided.
Added another reference for lines 210-212 (lines 175-177 in the prior version):
Li G, Kryczek I, Nam J, Li X, Li S, Li J, Wei S, Grove S, Vatan L, Zhou J, et al. LIMIT is an immunogenic lncRNA in cancer immunity and immunotherapy. Nat Cell Biol 2021;23:526-37.
------ references referred to above
Sell S. Stem Cell Origin of Cancer, AJP June 2010, Vol. 176, No. 6. DOI: 10.2353/ajpath.2010.091064
Marongiu, F., Serra, M. and Laconi, E. (2018) Development versus evolution in cancer biology. Trends in Cancer 4, 342–348, https://doi.org/10.1016/j.trecan.2018.03.007
Chen, H. and He, X.L. (2016) The convergent cancer evolution toward a single cellular destination. Mol. Biol. Evol. 33, 4–12, https://doi.org/10.1093/molbev/msv212
Aiello, N.M. and Stanger, B.Z. (2016) Echoes of the embryo: using the developmental biology toolkit to study cancer. Dis. Model. Mech. 9, 105–114, https://doi.org/10.1242/dmm.023184
Kim, D.H., Xing, T., Yang, Z., Dudek, R., Lu, Q. and Chen, Y.H. (2017) Epithelial mesenchymal transition in embryonic development, tissue repair and cancer: a comprehensive overview. J. Clin. Med. 7, 1, https://doi.org/10.3390/jcm7010001
Many thanks for suggesting the above references, which we have included in this manuscript.
Reviewer 2 Report
I have read the manuscript with interest, enthusiasm. Almost as I read a virtuous story. We do not often find scientific texts written as captivating narratives. From the first paragraph to the very last, every single sentence -and even word- has been carefully chosen, charmingly selected to provide the text with musicality and meaning. This review is not only magistral in the ideas and refreshing perspectives of an ancient question. It is also brilliant on the structure and its intrinsic poetry. Congratulations to the authors, and my most empathic recommendation for its prompt publication to the editors.
In this perspective review, the authors discuss about the convergences and divergences of the two main theories about cancer origin: the genetic versus the stem-based perspectives. They discuss about the simple and ancient questions “what is the origin of cancer” and “is cancer a somatic or a stem-based disease”. They defend the extraordinary resemblance between the oncogenesis and the ontogeny, and the common features of cancer cells and embryonic cells.
This discussion is not new and a remarkable proportion of the scientific community still refuses the idea of conceiving cancer as a stem-based disease. Despite the recent advances on cancer stem cell biology and behavior, experimental approaches that have completely shown new perspectives about cancer origin and causes of cancer heterogeneity, such as lineage tracing, suicide gene-based experiments, CRISPR-Cas9, organoids… there is still a general tabu about cancer stem cells and how relevant they are to explain most of the hallmarks of cancer. In this review, the authors explain, justify and reinforce this idea, elegantly and at the same time, disruptively.
The authors illustrate in this review that cancer is a multicellular rather than a unicellular process, and even more, a cellular rather than a genetic disease. They reveal that the incredible resemblance between a cancer cell and a stem cell suggests that they are intimately related. Such a unified theory can account for all cancer hallmarks, including metastasis, heterogeneity, dormancy and immune evasion. This vision of cancer origin predicts that integrated medicine could be more effective than precision medicine for the management of most tumors.
The authors divide the review in a number of chapters, accurately chosen, discussing about the similarities between oncology and ontogeny, how the former recapitulates the later; mini-organs frequently found in tumors; reminiscent embryogenesis detected in glioblastoma; the current and growing focus on neoantigens, and the possibility that multimodal therapy may be more beneficial than targeted therapy; the embryonic niche versus the stemness niche; the paradox of TGF and the EMT-MET; and the mutations in oncogenes and tumor suppressors and how those affect in different ways progenitor cells versus differentiated cells.
The chosen references support the discussion and endorse the debate of this review.
Figure 1 is certainly curious and appropriate for this particular review. Maybe Table 1 (summary of oncology recapitulating ontogeny) could be esthetically or artistically improved, to fit better in the global style of the review.
Author Response
We are more than impressed by the reviewer’s erudite review and humbled by his/her gracious comments of our manuscript! It is gratifying to know that there are more and more intrepid professionals with expertise and experience who dare to think outside the box and see beyond the horizon.
We have replaced the Table with another figure to contrast with the first figure and to further illustrate and emphasize the basic ideas of this paper, as recommended.
Reviewer 3 Report
Cancer is a heterogeneous structure consisting of differentiated and undifferentiated cells (stem cells). As the Authors emphasize, two theories of cancer formation are discussed in the literature and both are accepted. The role of stem cells in the development of cancer has been well documented.
it is not surprising to find stem cells in cancer tissue which are responsible for embryogenesis.
Generally the fetus is considered as allograft and antigenically immature foreign body, whereas, cancer is considered as autograft and immune response is different.
The Authors seems to just scratch the surface of the topic but this is not sufficient for publication.
The Authors spent a lot of time on speculative discussion rather than scientific evidence supporting the hypotheses put forward in the paper.
Author Response
A principal aim of this Perspective is to demonstrate that there is abundant evidence rather than mere hypothesis for a stem cell origin of cancers when it concerns tumorigenesis recapitulating embryogenesis.
It is true that a fetus and a cancer cannot be more different, but any similarity between the two is what may be missed or has been dismissed in our thinking. We propose that a similar biological process and mechanism of action between two disparate entities such as a fetus and a cancer in a stem cell origin and within the proper cellular context may provide a novel perspective in the way we understand cancer and how we treat it.
Importantly, we have provided updated research and recent evidence that support one rather than both contradictory theories of cancer formation, which the reviewer may have missed or chooses to dismiss. To assume that two opposing cancer theories are both acceptable may be prevalent but shows a lack of curiosity, which is common in conventional thinking and our current culture. Hopefully, this paper will dissuade such mentality and inclination.
Reviewer 4 Report
The manuscript by Tu Shi-Ming et al describes a very elaborate interpretation of cancer biology in which they considered the stem-cell versus genetic origin of cancer. Their narratives and perspectives on a malignant process focused on perpetuation of stem cell traits or stem-cell diseases, as examples the aberrant malignant teratocarcinomas that develop early during embryogenesis. Detailed genomic profiling of isolated cancer cells has confirmed how mutation in specific genes induces distinct hallmark capabilities and characteristics of cancer cells, allowing their selection and adaptation to growth in the primary and distant tissues. There are other complexes questions to be answered in their stem cell model as example, the changes in the metabolic pathways, which also seems to be dependent on specific mutation in cancer oncogenes. The examples presented in the manuscript support their propositions to demonstrate how stem cells without some specific genetic defects form malignant tumors. Finally, the manuscript is well written and presents one figure and one table that need to be rewritten to be more comprehensive to the readers.
Author Response
We thank the reviewers for your insightful comments and suggestions. We have provided our responses below in bold with changes highlighted in red underneath the reviewer comments. We have incorporated all changes recommended by the reviewers with the changes highlighted in red in the revised manuscript.
A principal aim of this Perspective is to demonstrate that there is abundant evidence rather than mere hypothesis for a stem cell origin of cancers when it concerns tumorigenesis recapitulating embryogenesis.
One way to reconcile the different opinions of the reviewers is to adopt a commentary style but adhere to the perspective structure of this article, as recommended. As a commentary, we will directly highlight the scientific evidence and recent publications (rather than just mention them in the references in case the reviewers may not have read or noticed them)! As a perspective, we would like to preserve the details and comprehensiveness which the reviewers expect and demand for this article.
Thank you for your astute and insightful comments! We completely agree with you that a unified theory and a stem cell origin of cancer need to encompass genomics, epigenomics, metabolomics, etc. of cancer (references 1 and 2). It needs to embrace the various compartments, components, and the microenvironment of cancer.
We have replaced the Table with another figure to contrast with the first figure and to further illustrate and emphasize the basic ideas of this paper, as recommended.
Round 2
Reviewer 1 Report
Although the authors responded to my comments in their rebuttal, I do feel that some of the issues are still not appropriately reflected in the manuscript itself.
Specifically:
- The authors agree that regarding the stem-cell theory, “an introduction to this subject may clarify the questions” I asked. Still, the introduction does not provide any background on this theory (basically, what does the theory states), and what are the issues that the author feel need addressing... As a response, they indicate that the introduction cites two of their own references (including a book with a large number of references) discussing the stem-cell theory... So, the reader will have to read those references first. The first time the concept of the stem-cell theory of cancer is mentioned is in the section of “Oncology recapitulates ontology”: “This pivotal observation is key to unlocking a stem-cell theory of cancer. It is fundamental to the idea that cancer is a stem-cell disease [1,2]. “ Again, it is not clear what “a stem-cell theory of cancer” refers to (no references are provided; is this a new theory that authors want to develop? Hence the “a” – ie, new, instead of “the”- ie, already proposed); and what it is meant by “unlocking”? Later, in the section “Mini-organs” the “stem-cell theory” is contrasted with the genetic theory; but again, there are no references for this theory: “In contrast, the stem-cell theory of cancer hypothesizes that aberrant stem-cells with or without genetic mutations are the source, if not cause, of heterogeneity in cancer. “... And the next sentence cites again two papers of the authors: “The latter theory predicts that those same genetic defects occurring in a differentiated cell will be nonmalignant, and in a progenitor cell with less stem-ness features less malignant with less heterogeneity [11,12]. “
- Regarding the concept “oncology recapitulates ontology”, I agree with the authors view that that practice of oncology is a process; but can’t see how the practice of oncology recapitulates ontology... of course, the practice of oncology would benefit from understanding the potential links of cancer to ontology, but that is not what “oncology recapitulates ontology” implies...
- To somewhat address the point above, the authors changed the title to include the point about Cancer Care (ie, practice of oncology). However, the Abstract does not make any reference to the importance of understanding the links between tumorigenesis and embryogenesis for cancer treatment (which I think is the main novel aspect of this manuscript).
- I still think the use of “recapitulating” in either concepts – “oncology recapitulates ontology” or “tumorigenesis recapitulates embryogenesis”, is not appropriate. In two instances in the manuscript, the authors use the term “recapturing” ie., “tumorigenesis recapturing embryogenesis”; I think that “recapture” is a better word for this statement.
In their response, the authors agree that the connection between embryogenesis and carcinogenesis is well known: “Agree with reviewer about the well-known connection between developmental processes and carcinogenesis, which is the main theme of this article”. The authors then should be more specific as to what is their contribution and highlight the novelty of this manuscript.
-
Author Response
We thank the reviewers for your insightful comments and suggestions. We have provided our responses below in bold with changes highlighted in red underneath the reviewer comments. We have incorporated all changes recommended by the reviewers with the changes highlighted in red in the manuscript.
Reviewer 1
Although the authors responded to my comments in their rebuttal, I do feel that some of the issues are still not appropriately reflected in the manuscript itself.
Specifically:
- The authors agree that regarding the stem-cell theory, “an introduction to this subject may clarify the questions” I asked. Still, the introduction does not provide any background on this theory (basically, what does the theory states), and what are the issues that the author feel need addressing... As a response, they indicate that the introduction cites two of their own references (including a book with a large number of references) discussing the stem-cell theory... So, the reader will have to read those references first. The first time the concept of the stem-cell theory of cancer is mentioned is in the section of “Oncology recapitulates ontology”: “This pivotal observation is key to unlocking a stem-cell theory of cancer. It is fundamental to the idea that cancer is a stem-cell disease [1,2]. “ Again, it is not clear what “a stem-cell theory of cancer” refers to (no references are provided; is this a new theory that authors want to develop? Hence the “a” – ie, new, instead of “the”- ie, already proposed); and what it is meant by “unlocking”? Later, in the section “Mini-organs” the “stem-cell theory” is contrasted with the genetic theory; but again, there are no references for this theory: “In contrast, the stem-cell theory of cancer hypothesizes that aberrant stem-cells with or without genetic mutations are the source, if not cause, of heterogeneity in cancer. “... And the next sentence cites again two papers of the authors: “The latter theory predicts that those same genetic defects occurring in a differentiated cell will be nonmalignant, and in a progenitor cell with less stem-ness features less malignant with less heterogeneity [11,12]. “
Appreciate very much the reviewer’s excellent suggestions to strengthen our introduction by adding statements (paragraph 4, sentences 2-3) with appropriate references about the background of the stem-cell theory and the issues we would like to address in this manuscript.
We elaborate that the current concepts of a stem-cell theory of cancer [1,2] reiterates the conventional ideas of tumorigenesis recapitulating embryogenesis. We demonstrate that a proper cancer theory on a stem-cell versus genetic origin of cancers [1,2] entails pivotal implications in cancer research and for cancer care.
- Regarding the concept “oncology recapitulates ontology”, I agree with the authors view that that practice of oncology is a process; but can’t see how the practice of oncology recapitulates ontology... of course, the practice of oncology would benefit from understanding the potential links of cancer to ontology, but that is not what “oncology recapitulates ontology” implies...
We have changed “recapitulating” to “recapturing” to better depict the potential links of cancer to ontology in the Abstract (paragraph 1), introduction (paragraph 3), sections 1 (title and last sentence); 8 (paragraph 5); and Conclusions (paragraph 4), as recommended.
- To somewhat address the point above, the authors changed the title to include the point about Cancer Care (ie, practice of oncology). However, the Abstract does not make any reference to the importance of understanding the links between tumorigenesis and embryogenesis for cancer treatment (which I think is the main novel aspect of this manuscript).
Agree with reviewer. We have added a statement about importance of understanding the link between tumorigenesis and embryogenesis for cancer treatment in the Abstract, paragraph 5.
Importantly, targeting stem-like pathways has therapeutic implications, because stem-ness may be the true driver, if not engine, of the malignant process. Furthermore, anti-stem-like activity elicits anti-cancer effects for a variety of cancers, because stem-ness features may be a universal property of cancer.
- I still think the use of “recapitulating” in either concepts – “oncology recapitulates ontology” or “tumorigenesis recapitulates embryogenesis”, is not appropriate. In two instances in the manuscript, the authors use the term “recapturing” ie., “tumorigenesis recapturing embryogenesis”; I think that “recapture” is a better word for this statement.
We have changed “recapitulating” to “recapturing” to better depict the potential links of cancer to ontology in the Abstract (paragraph 1), introduction (paragraph 3), sections 1 (title and last sentence); 8 (paragraph 5); and Conclusions (paragraph 4), as recommended.
In their response, the authors agree that the connection between embryogenesis and carcinogenesis is well known: “Agree with reviewer about the well-known connection between developmental processes and carcinogenesis, which is the main theme of this article”. The authors then should be more specific as to what is their contribution and highlight the novelty of this manuscript.
Many thanks to the reviewer for pointing out that connecting tumorigenesis and embryogenesis with cancer treatment could be the main novel aspect of this manuscript.
We have added a new section to provide specific examples with references to highlight this link and emphasize the novelty of this article:
- Therapeutic vs Theoretic
It is important to separate practice from principles and reality from theories in cancer care. However, ideas and actions are often inextricably linked: what we believe affects what we do. Different cancer theories will lead to distinct research directions and treatment developments with disparate clinical implications.
Perhaps it is not mere coincidence that anti-stem-ness activity has anti-cancer effects, because of an intrinsic yet intricate, parallel but duplicate interplay between embryogenesis and oncogenesis, and between stem cell and cancer biology.
Trop2
Trop2 was first isolated from monoclonal antibodies generated against a human choriocarcinoma cell line BeWo, as well as from nonmalignant trophoblast cells [64]. It is highly expressed in stem cells within various organs during embryogenesis, and in a variety of human malignancies, including triple negative breast cancer (TNBC) and upper tract urothelial carcinoma. Over-expression of Trop2 in human tumors promotes tumor aggressiveness and increases patient mortality [65,66].
Trop2’s multifaceted role includes regulation of cell proliferation and migration, self-renewal, and maintenance of basement membrane integrity. It enhances stem-like properties of cancer cells through beta-catenin signaling [67].
Sacituzumab-govitecan is an effective cancer therapeutic that targets Trop2. It is approved for the treatment of refractory TNBC and metastatic urothelial carcinoma [68,69].
Nectin4
Among the 4 human nectins, nectin4 is unique in that its expression is largely restricted to placental and embryonic tissues. In contrast to healthy adult tissue, many cancer types, including breast, ovarian, cervical, colorectal, esophageal, gastric, lung, liver, and thyroid cancers, have high nectin4 expression. High nectin4 expression is associated with increased tumor size, grade, and invasiveness, as well as reduced patient survival [70].
Nectin4 is a stem-cell biomarker that upregulates EMT and metastasis. It induces WNT/beta-catenin signaling via Pi3k/Akt axis [71]. It cooperatively regulates with p95-ErbB2 Hippo signaling-dependent SOX2 expression [72].
Enfortumab is an effective cancer therapeutic that targets nectin4. It is approved for the treatment of locally advanced or metastatic bladder cancer [73].
RTK
Receptor tyrosine kinases (RTK) are ubiquitous stem-ness factors that play a vital role in diverse embryonic and malignant processes, including pluripotency/differentiation, self-renewal/cell fate, morphogenesis, migration/invasion, et cetera [74,75].
Tyrosine kinase inhibitors (TKI) are known to be teratogenic, in part because they disrupt embryogenesis; Not unexpectedly, because TKI also interfere with oncogenesis they can be utilized for therapeutic purposes in cancer care.
For example, HER-2 is innately connected with stemness. It interacts with IL-8R (CXCR1/2) in the regulation of breast CTC by the tumor microenvironment [76,77].
Several HER-2 targeted therapies are effective and have been approved for the treatment of HER-2+ breast and gastric cancers [78].
- Lipinski M, Parks DR, Rouse RV, Herzenberg LA. Human trophoblast cell-surface antigens defined by monoclonal antibodies. Proc Natl Acad Sci USA 1981; 78:5147-50.
- McDougall ARA, Tolcos M, Hooper SB, Cole TJ, Wallace MJ. Trop2: from development to disease. Dev Dyn 2015; 244:99-109.
- Shvartsur A, Bonavida B. Trop2 and its overexpression in cancers: regulation and clinical/therapeutic implications. Genes & Cancer 2015;6: 84-105.
- Stoyanova T, Goldstein AS, Cai H, Drake JM, Huang J, Witte ON. Regulated proteolysis of Trop2 drives epithelial hyperplasia and stem cell self-renewal via beta-catenin signaling. Genes Dev 2012;26:2271-85.
- Bardia A, Mayer IA, Vahdat LT, Tolaney SM, Isakoff SJ, Diamond JR, O’Shaughnessy J, Moroose RL, Santin AD, Abramson VG, et al. Sacituzumab govitecan-hziy in refractory metastatic triple-negative breast cancer". N Engl J Med 2019; 380:741–751.
- Tagawa ST, Balar AV, Petrylak DP, Kalebasty AR, Loriot Y, Flechon A, Jain RK, Agarwal N, Bupathi M, Barthelemy P, et al. TROPHY-U-01: a phase II open-label study of Sacituzumab govitecan in patients with metastatic urothelial carcinoma progressing after platinum-based chemotherapy and checkpoint inhibitors. J Clin Oncol 2021; 39:2474-85.
70 Chatterjee S, Sinha S, Kundu CN. Nectin cell adhesion molecule-4 (NECTIN-4): a potential target for cancer therapy. Eur J Pharmacol 2021;911:174516.
- Siddarth S, Goutam K, Das S, Nayak A, Nayak D, Sethy C, Wyatt MD, Kundu CN. Nectin-4 is a breast cancer stem cell marker that induces WNT/beta-catenin signaling via Pi3k/Akt axis. Int J Biochem Cell Biol 2017; 89:85-94.
- Kedashiro S, Kameyama T, Mizutani K, Takai Y. Nectin-4 and p95-erbB2 cooperatively regulate Hippo signaling-dependent SOX2 gene expression, enhancing anchorage-independent T47D cell proliferation. Sci Rep 2021; 11:7344.
- Powles T, Rosenberg JE, Sonpavde GP, Loriot Y, Duran I, Lee JL, Matsubara N, Vulsteke C, Castellano D, Wu C, et al. Enfortumab vedotin in previously treated advanced urothelial carcinoma. N Engl J Med 2021; 384:1125-1135.
- Mele S, Johnson TK. Receptor tyrosine kinases in development: insights from Drosophila. Int J Mol Sci 2020; 21:188.
- Chen J, Song W, Amato K. Eph receptor tyrosine kinases in cancer stem cells. Cytokine Growth Factor Rev 2015; 26:1-6.
- Duru N, Fan M, Candas D, Menaa C, Liu HC, Nantajit D, Wen Y, Xiao K, Eldridge A, Chromy BA, et al. HER2-associated radioresistance of breast cancer stem cells isolated from HER2-negative breast cancer cells. Clin Cancer Res 2012;18:6634-47.
- Korkaya H, Kim GI, Davis A, Malik F, Henry NL, Ithimakin S, Quraishi AA, Tawakkol N, D’Angelo R, Paulson AK, et al. Activation of an IL6 inflammatory loop mediates trastuzumab resistance in HER2+ breast cancer by expanding the cancer stem cell population. Mol Cell 2012;47:570-84.
- Oh DY, Bang YJ. HER2-targeted therapies – a role beyond breast cancer. Nat Rev Clin Oncol 2020; 17: 33-48.
Reviewer 3 Report
In my opinion the characteristics of stem cells and cancer stem cells are poorly described. It would be interesting to show the similarities and differences between stem cells and cancer stem cells.
Author Response
We thank the reviewers for your insightful comments and suggestions. We have provided our responses below in bold with changes highlighted in red underneath the reviewer comments. We have incorporated all changes recommended by the reviewers with the changes highlighted in red in the manuscript.
Reviewer 3
In my opinion the characteristics of stem cells and cancer stem cells are poorly described. It would be interesting to show the similarities and differences between stem cells and cancer stem cells.
Agree with reviewer that further research to investigate the similarities and differences between normal stem cells and cancer stem cells will be critical in our understanding about the origin and nature of cancer as well as for drug vs therapy development in cancer care. Hopefully, this manuscript will inspire and instigate scientists and clinicians toward this goal.